# Breastfeeding and Prevalence of Metabolic Syndrome among Perimenopausal Women

**DOI:** 10.3390/nu12092691

**Published:** 2020-09-03

**Authors:** Edyta Suliga, Elzbieta Ciesla, Martyna Gluszek-Osuch, Malgorzata Lysek-Gladysinska, Iwona Wawrzycka, Stanislaw Gluszek

**Affiliations:** 1Institute of Health Sciences, The Jan Kochanowski University in Kielce, ul. Zeromskiego 5, 25-369 Kielce, Poland; eciesla@ujk.edu.pl (E.C.); mgluszekosuch@ujk.edu.pl (M.G.-O.); iwona.wawrzycka@ujk.edu.pl (I.W.); 2Institute of Biology, The Jan Kochanowski University in Kielce, ul. Zeromskiego 5, 25-369 Kielce, Poland; malgorzata.lysek-gladysinska@ujk.edu.pl; 3Institute of Medical Sciences, The Jan Kochanowski University in Kielce, ul. Zeromskiego 5, 25-369 Kielce, Poland; sgluszek@wp.pl

**Keywords:** lactation, parity, metabolic health

## Abstract

Little is known about the long-term benefits of breastfeeding for mother’s metabolic health. This study aimed to investigate the links between breastfeeding duration and the prevalence of metabolic syndrome (MetS) and its components in perimenopausal women. The analysis included a group of 7621 women aged 55.4 ± 5.4 years. MetS and its components were defined according to the International Diabetes Federation guidelines. Women who breastfed for 13–18 months and beyond 18 months were at lower risk of MetS (odds ratio OR) = 0.76, 95% CI 0.60–0.95; *p* = 0.017 and OR = 0.79, 95% CI 0.64–0.98; *p* = 0.030, respectively) than those who never breastfed. Meanwhile, women who breastfed for 7–12 months showed increased glucose concentration (OR = 0.77, 95% CI 0.63–0.94; *p* = 0.012) compared with those who had never breastfed. The additional analysis involving parity showed that women who had given birth to two babies and breastfed them had lower odds of MetS than those who never breastfed (*p* < 0.05), although there was no significant difference among women who breastfed for >18 months. Women who had given birth to at least three children and breastfed for 1–6 and 13–18 months had lower odds of MetS and increased triglyceride concentration (*p* < 0.05). Moreover, participants having breastfed for 1–6 months were found to have a reduced risk of abdominal obesity compared with those who had not breastfed (*p* < 0.05). Breastfeeding is associated with lower prevalence of MetS in perimenopausal women and can be recommended as a way of reducing the risk of MetS and its components.

## 1. Introduction

Women undergo hormonally regulated metabolic changes during pregnancy to ensure a supply of adequate substrates to the developing fetus. In the second and third trimester of pregnancy, physiological hyperinsulinemia and insulin resistance can occur, resulting in higher glucose and free fatty acid levels in the mother [1,2]. Moreover, an atherogenic lipid profile develops, i.e., total cholesterol, low-density lipoprotein (LDL) cholesterol, high-density lipoprotein (HDL) cholesterol, and lipoprotein (a) concentrations increase [1,3]. These metabolic changes usually normalize within a few months after birth [4], but can persist for longer in some women, increasing the risk of type 2 diabetes and cardiovascular diseases in future life [5,6,7,8].

Studies indicate that the improvement in metabolic parameters can occur more quickly and completely during breastfeeding [9]. Indeed, lower blood glucose concentrations, as well as improved insulin sensitivity and lipid profiles, were reported among breastfeeding mothers examined several months after childbirth [10,11,12]. Despite this, the potential long-term benefits of lactation in terms of maternal metabolic health are poorly understood [13,14]. Perrin et al. found breastfeeding had a beneficial effect on the risk of diabetes, hypertension, and cardiovascular diseases, with slightly less evidence concerning weight gain and dyslipidemia [13]. Nguyen et al. demonstrated a dose–response relationship between lactation duration and the prevalence of metabolic risk factors and metabolic syndrome (MetS), but they also noted that this relationship should be interpreted with caution because the evidence comes from a small number of observational studies [14]. In addition, postmenopausal women who lactated for >12 months were less likely to have hypertension than those who never breastfed [15]. Moreover, women aged ≤50 years who had breastfed for >24 months during their lifetime were found to have 5% lower total cholesterol concentration and 17% lower triglyceride (TG) concentration compared with women of the same age who had never breastfed [16]. Furthermore, a prospective study showed that breastfeeding for more than 6 months was significantly associated with a reduced risk of developing MetS compared with lactation lasting 0–1 month [17]. However, a recent cohort study did not confirm a positive relationship between longer lactation duration and improved maternal metabolic health after 12 years of follow-up [18]. Therefore, the effect of lactation duration on maternal metabolic health remains inconclusive.

This study aimed to investigate the associations between breastfeeding duration and the prevalence of MetS and its components in perimenopausal women. All analyses were performed in the entire study group, and separately, in three parity categories. Additionally, the analyses of the above relationships were carried out taking into account the category of parity. The likelihood of the MetS development in women who had ever breastfed (regardless of the duration of lactation) was also examined compared to women who gave birth to at least 1 child but never breastfed. Moreover, the prevalence of MetS in parous vs. nulliparous women was analyzed.

## 2. Materials and Methods

### 2.1. Study Participants

Data from female participants enrolled in the POlish-Norwegian Study (PONS) project, were obtained. The PONS project was a longitudinal observational regional research on the health status and prevalence of chronic non-communicable diseases in south-eastern Poland. In this study, we used cross-sectional data from the first edition of the PONS project. In this study participated 13,172 volunteers from the city of Kielce and surrounding rural area (Świętokrzyskie Province) in Poland, aged 37–66 years. Among the 8725 female study participants, 327 women diagnosed with cancer and 777 women with missing data were excluded. Finally, data from 7621 women were included in the analyses. The sample size used in this study was large enough to detect any statistical differences. The mean age of participants was 55.4 ± 5.4 years.

### 2.2. Bioethics Committee Approval

The study was approved by the Committee on Bioethics at the Faculty of Health Sciences, Jan Kochanowski University in Kielce, Poland (No. 29/2015) (data analysis).

### 2.3. Anthropometric Measurements and Biochemical Profiling

Basic anthropometric measurements and biochemical analysis of blood were carried out. The results of height and body weight measurements were used to calculate the body mass index (BMI), and the waist circumference was used to determine abdominal obesity. Blood pressure was measured in duplicate and the mean of both measurements was used in the analysis. The concentration of triglycerides was measured using the enzymatic method, with phosphoglycerol oxidase and determination of H_2_O_2_ (with peroxidase). The concentration of HDL-cholesterol was obtained using the colorimetric non-precipitation method with polyethylene glycol modified enzymes. The glucose concentration in the blood serum was assessed using the enzyme method with hexokinase. Measurements and biochemical profiles of project participants were described in detail in the previous articles [19,20]. MetS and its components were defined according to the International Diabetes Federation (IDF) Task Force on Epidemiology and Prevention guidelines [21] in the presence of 3 or more of the following 5 risk factors: waist circumference ≥80 cm, elevated fasting glucose (≥100 mg/dL or drug treatment of elevated glucose or previously diagnosed type 2 diabetes), elevated blood pressure (BP) (systolic BP ≥130 and/or diastolic BP ≥85 mm Hg or antihypertensive drug treatment or previously diagnosed hypertension), hypertriglyceridemia (≥150 mg/dL or drug treatment for elevated triglycerides, and reduced HDL-cholesterol (<50 mg/dL), or drug treatment for reduced HDL-cholesterol or previously diagnosed dyslipidemia.

### 2.4. Sociodemographic and Lifestyle Data

Sociodemographic data (age, menopausal status, education, place of residence, parity) and information on lifestyle: breastfeeding, total (lifetime) duration of breastfeeding, smoking, and the use of hormone therapy, were gathered in the survey. The division into breastfeeding categories was made on the basis of full months of feeding (never, 1–6, 7–12, 13–18, >18 months). The International Physical Activity Questionnaire (IPAQ)—Long Form was used to assess physical activity and sitting time [22]. Based on data obtained through questionnaires, total physical activity, expressed as metabolic equivalents (MET/min/week^−1^), and sitting time (minutes/day) were calculated. It was not feasible to calculate the caloric values of participants’ diets based on data on food intake collected with a Food Frequency Questionnaire. Therefore, the dietary data were classified in the analysis into three dietary patterns (DPs), previously identified in this population (i.e., healthy, traditional-carbohydrate, and westernized). A detailed description of dietary patterns and the procedures for their development were previously published [23].

### 2.5. Dependent, Independent, and Confounding Variables

The dependent variables (categorical) include metabolic syndrome (yes—1, no—0) and its five components: abdominal obesity (yes—1, no—0), increased glucose concentration (yes—1, no—0), elevated blood pressure (yes—1, no—0), increased triglyceride concentration (yes—1, no—0), and decreased HDL-cholesterol concentration (yes—1, no—0).

Depending on the statistical analysis, the following independent variables were included: breastfeeding status (ever-1, never-0), and breastfeeding duration in all group of women, who gave birth (never, breastfed: 1–6, 7–12, 13–18, and more than 18 months). The analyses were then performed for parous women. The independent variables were parity (parous-1, nulliparous-0) and breastfeeding duration analyzed in three parity categories (1, 2, 3, and more births)—in parous women. These were qualitative, categorical variables.

Confounding variables were: age (calendar years completed during research), years of education (number of years spent in schools), physical activity (sum of moderate, vigorous, and walking physical activity expressed in METs/min/day^−1^), sitting time (time spent in sitting or lying position excluding sleep expressed in min/day), and BMI (kg/m^2^), as continuous variables. Place of living (city: a settlement unit with Polish municipal rights—1, village: settlement unit without municipal rights of agricultural character—0), marital status (married or in a stable relationship—1, single: unmarried, widow/widower, divorced, or separated—0), parity (0, 1, 2, 3, and more births), menopausal status (premenopausal: still menstruating or not menstruating for less than 12 months—0, postmenopausal: women with amenorrhea for at least 12 months—1), hormone therapy (yes: women who had ever used hormone therapy—1, no: women who had never used hormone therapy—0), smoking (yes: ever smoked—1, no: never smoked—0), traditional-carbohydrate, healthy, and westernized dietary patterns (divided by tertiles: T1–0, T2–1, T3–2). These were qualitative, categorized variables. The above-mentioned confounding factor was included in the analysis due to their relationship with the prevalence of MetS and its components, confirmed in previous studies [17,23,24,25,26].

### 2.6. Statistical Analysis

For all continuous, confounding variables (age, years of education, physical activity, sitting time, and BMI), arithmetic means (X) and standard deviations (SD) were calculated. The t-test was applied to compare two independent arithmetic means in groups of the metabolic syndrome and its components (Table 1). A similar procedure was used in the presentation of these variables in the nulliparous and parous women groups (Appendix A). In the case of larger number of means in the analysis (i.e., the category of total duration of breastfeeding), one-way analysis of variance (ANOVA) was applied, and post-hoc Bonferroni tests were carried out to compare pairs of means (Appendix A). Qualitative variables (categorical from the group of independent, dependent and confounding variables) were expressed as frequency (N) and percentage (%). Differences in basic characteristics between participants with and without MetS and its components were assessed using the Chi-squared test for categorical variables (Appendix A).

The incidence of MetS and its components in relation to the duration of breastfeeding was evaluated with multivariate logistic regression analyses. The odds ratios (ORs) and 95% confidence intervals (CIs) were calculated for breastfeeding status, and for the following categories of feeding duration: never, 1–6, 7–12, 13–18, >18 months. The reference group consisted of women who gave birth but had never breastfed. The analyses were then performed for parous women (Appendix A).

In adjusted models, the following variables were considered confounding: continuous variables including age, years of education, physical activity, sitting time, dietary patterns, and body mass index, and categorical variables including place of living, marital status, menopausal status, smoking, hormone therapy, and parity (Table 2. The next adjusted model the influence of breastfeeding duration on MetS and its components in groups of parity were presented (Table 3. Analyses were performed using the statistical package Statistica, version 13.3 (TIBCO SOFTWARE CO. PL) and Plus set version 5.0.85 (STATSOFT POLSKA SP. Z.O.O. 2017). The significance level was set at *p* < 0.05.

## 3. Results

### 3.1. Characteristics of MetS, Breastfeeding Duration, Sociodemographic, Biomedical, and Lifestyle Factors of the Study Participants

Participants diagnosed with MetS or any of its components were older, more frequently postmenopausal, less educated, less physically active, had greater BMI, and were less likely to use hormone therapy compared with women without MetS and its components (Table 1). Women with MetS were less likely to be married or in a permanent relationship, gave birth to more children, were found to spend less time sitting, and had low adherence to a “healthy” dietary pattern (DP) compared with women without MetS.

Among participants who displayed abdominal obesity, there were more rural than urban women compared with women without abdominal obesity. Moreover, married women, those who gave birth to three or more children, women who reported having breastfed for longer (total >6 months), and a shorter sitting time predominated in the group of participants with abdominal obesity. There was also a lower proportion of smokers in this group than in women without abdominal obesity. However, there were no differences between the groups in terms of dietary patterns or the level of physical activity.

Participants with increased glucose concentration were more often multiparous and reported slightly shorter sitting time than women with normal glucose concentration. In the group of participants with increased TG concentration, there were fewer married women, more smokers, and a higher proportion of those who had never breastfed. Women with elevated blood pressure (BP) and dyslipidemia were more likely to have a traditional-carbohydrate DP and lower adherence to a healthy DP. The prevalence of MetS and its components did not vary depending on the score on the westernized DP.

Compared with nulliparous women, parous women were younger, less educated, more often lived in rural areas, and were married, more frequently used hormone therapy, had a lower score on westernized DP, were more physically active, and had a shorter sitting time. However, parous women were more likely to display abdominal obesity and have a higher mean BMI (Appendix A). No differences were found among the two groups regarding menopausal status, the proportion of smokers, adherence to the Traditional-carbohydrate DP, the occurrence of elevated blood pressure, abnormal glucose concentration, or dyslipidemia.

Participants who had never breastfed were older and more frequently postmenopausal than those who breastfed (Appendix A). Women who had breastfed for the longest periods (>18 months cumulatively) had the lowest level of education, more often lived in rural areas, and were married, gave birth to the highest number of children, were less likely to smoke, and used hormone therapy more often. As the participants’ total duration of breastfeeding lengthened, better adherence to the westernized DP and increased physical activity was reported. However, longer breastfeeding duration was associated with a higher mean BMI and a higher prevalence of abdominal obesity. Women who had never breastfed had the highest prevalence of MetS (46.81%) and increased TG concentration (34.16%). Nonetheless, no differences were found among the groups regarding sitting time, scores on the healthy and traditional-carbohydrate DPs, the prevalence of abnormal glucose or HDL concentrations, or elevated blood pressure.

### 3.2. Analysis of the Relationship between Breastfeeding Duration and the Presence of MetS and Its Components

Analysis of unadjusted ORs showed that women who breastfed had a lower incidence of MetS and increased glucose and TG concentrations compared with those who had never breastfed (Appendix A). Inclusion of lactation duration in the analysis confirmed significantly lower odds of MetS and increased TG concentration in breastfeeding women (except for those who had breastfed for 7–12 months). Lower risk of increased glucose concentration was observed in participants who had breastfed for 1–12 months than in those who had not breastfed. Moreover, women who had breastfed for 1–6 months had a lower risk of abdominal obesity than those who had never breastfed.

After taking confounding variables into consideration, lower ORs for MetS and increased glucose and TG concentrations were reported for women who had breastfed than those who had never breastfed, albeit not significantly (Table 2). Compared to women who had never breastfed, those who breastfed for 13–18 months and beyond 18 months had a lower risk of MetS (by 24% and 21%, respectively), and those who breastfed for 7–12 months had a lower risk of increased glucose concentration (by 23%).

### 3.3. The Probability of Development of MetS and Its Components in Relation to Parity and Breastfeeding Duration

Analysis of unadjusted ORs for parity groups demonstrated that parous women were more likely to develop abdominal obesity than nulliparous women (Appendix A). Participants who had given birth to one child and breastfed for 13–18 months had a lower risk of increased TG concentration than those who never breastfed. Meanwhile, those who had given birth to two children and breastfed for >18 months had a lower risk of MetS compared with women who had never breastfed, although the difference was not significant. Women who had given birth to three or more children and breastfed for 1–6 months had a lower risk of MetS, abdominal obesity, and increased TG concentration than those who never breastfed. In addition, those who had breastfed for 13–18 months and >18 months and who had given birth to three or more children had a lower risk of MetS and those who breastfed for >18 months had also a lower risk of increased glucose concentration.

After taking confounding variables into consideration, a lower risk of elevated BP was reported for parous women than nulliparous women (Table 3). No relationships were found between lactation duration and the risk of MetS and its components in women with only one child. Participants who had given birth to two children and breastfed them had lower odds of MetS than those who had never breastfed (although the OR in women who breastfed >18 months did not achieve statistical significance). Women who had given birth to three or more children and breastfed for 1–6 months or 13–18 months had lower odds of MetS and increased TG concentration. Furthermore, the risk of abdominal obesity was reduced in participants who breastfed for 1–6 months compared with those who had never breastfed.

## 4. Discussion

We found that women who had previously breastfed were less likely to develop MetS and increased TG and glucose concentrations than those who had never breastfed. After adjusting for confounding variables, the relationships became weaker for the general study population and did not reach statistical significance. However, analysis in four categories of lactation duration demonstrated that more than 12 months of breastfeeding significantly reduced the odds of MetS.

Our findings agree with the results of many long-term [17,24] and cross-sectional studies [25,26,27], indicating a lower prevalence of MetS in women who breastfed, and the differences in the compared results concern only the duration of lactation, after which such an effect was observed. For example, Gunderson et al. showed that among women of childbearing age observed for 20 years, a lactation duration of >5 months was significantly associated with a lower incidence of MetS compared with lactation lasting from 0 to 1 month [17]. Similarly, during a 9-year follow-up, Tehrani et al. reported that lactation duration protected women from MetS in a dose–response manner, with the strongest protective effect observed in women who breastfed ≥24 months [24]. Moreover, a cross-sectional study of over 2500 middle-aged, parous women of different ethnicity also found that breastfeeding was associated with a lower prevalence of MetS in a dose–response manner [25]. Breastfeeding was also linked to a lower prevalence of MetS in women aged ≥20 years in a study by Cohen et al. [26]. They found, however, that weight changes can influence these associations between breastfeeding and the risk of MetS, as the relationships were no longer significant after adjusting for BMI. In contrast, in our study, we found that even after including BMI, some relationships between breastfeeding and MetS risk remained significant. Meanwhile, no association between lactation history and MetS was found in a study involving 892 Korean postmenopausal women [27].

Further analysis of unadjusted ORs showed that the prevalence of increased glucose and TG concentrations was lower in participants who had breastfed than among those who had never breastfed. In addition, the inclusion of lactation duration demonstrated that women who breastfed had a significantly lower incidence of increased TG concentration in three of the four categories of total breastfeeding duration. After adjusting for confounders, only a reduced risk of increased glucose concentration in women having breastfed for 7–12 months remained significant.

Other authors’ findings on the relationship between breastfeeding and the occurrence of single metabolic disorders that are components of MetS vary. For example, Ram et al. demonstrated that after adjusting for confounding variables, lactation reduced the risk of abdominal obesity and elevated blood pressure [25]. However, they did not find a significant relationship between lactation and glucose, TG, or HDL concentrations. Nevertheless, Schwarz et al. proved that postmenopausal women who reported a lifetime history of lactation of ≥12 months were less likely to have hypertension or hyperlipidemia compared with women who had never breastfed [15]. In addition, women aged ≤50 years whose lifetime duration of breastfeeding was beyond 24 months had lower TG concentrations than women of the same age who had never breastfed [16]. A study by Wiklund et al. among women aged 16–20 years following pregnancy demonstrated that those who had breastfed for <6 months had higher body fat mass, especially in the abdominal region, higher TG, and fasting serum glucose concentrations, as well as higher blood pressure than women who had breastfed for longer [28]. Ra and Kim showed that in postmenopausal women, breastfeeding was not significantly associated with the presence of MetS, but decreased the likelihood of abdominal obesity [29]. Moreover, a meta-analysis of studies including over 200,000 women showed that breastfeeding for at least 12 months was associated with a relative risk reduction of 30% for type 2 diabetes and 13% for hypertension [30].

In contrast, several recent studies have not confirmed the association between lactation and the occurrence of single metabolic risk factors in the long term. During a 12-year follow-up, Velle-Forbody et al. found no beneficial effect of long breastfeeding duration on serum glucose and lipid concentrations, as well as waist circumference or blood pressure [18]. Similarly, a study among Brazilian women from the Pelotas Birth Cohort did not find any relevant associations between breastfeeding and cardiovascular risk factors [31]. Many authors emphasize that other factors, apart from lactation, are more likely to affect women’s metabolic health, including parity.

Previous studies show that the probability of obesity increases with each successive birth [32,33,34]. Among the participants of our study, a higher risk of abdominal obesity (waist circumference ≥80 cm) was also observed in parous women, compared with nulliparous ones, although only in the unadjusted model. Some studies also indicate that parous women may have a more atherogenic lipid profile than nulliparous, as well as a higher risk of coronary heart disease in later life [34,35,36], which was not supported in our population. Parity proved to be a factor that strongly modified the associations between breastfeeding and the prevalence of MetS. Additional analysis in three parity categories showed that after the inclusion of confounders, there was no relationship between lactation duration and the prevalence of MetS and its components among participants with only one child. However, this finding may have resulted from the shorter lactation duration observed in this group of women compared with women who gave birth to more children. Indeed, in women who had given birth to two children, the mean breastfeeding duration was associated with a lower risk of MetS, while those who had given birth to three or more children had a lower risk of MetS and increased TG concentration. Women who breastfed for 1–6 months also had lower odds of abdominal obesity.

The mechanism underlying the effect of lactation on maternal metabolic health, which is observed over many years after the cessation of lactation, remains unexplained. It is known that breastfeeding mobilizes fat stores that accumulate during pregnancy [9]. Increased use of body fat stores, especially visceral fat [37,38], can be associated with long-term improvements in metabolic profile. Maternal metabolism is also affected by hormone changes; for example, prolactin plays a vital role in regulating insulin secretion and glucose homeostasis, and prolactin levels increase during lactation [39]. Furthermore, Much et al. found that >3 months of lactation affects changes in some metabolites (especially phospholipids and branched-chain amino acids), and thus, can correct disruptions of metabolic pathways involved in, among others, early pathogenesis of type 2 diabetes [40]. The authors emphasize that these beneficial changes persist for several years post-weaning.

### Strengths and Limitations

The major limitation, which may have affected the results of this study, lies in the fact that certain health behaviors tend to co-exist [41,42,43,44]. Women who decided to breastfeed might have led a healthy lifestyle in general. However, the inclusion of numerous sociodemographic and lifestyle variables in this study significantly reduced their confounding influence on the analyzed relationships. Moreover, we found the inclusion of these variables did not significantly influence the results concerning the relationships. Another limitation was the lack of information on exclusivity and intensity of breastfeeding, as well as the retrospective evaluation of lactation duration. However, Promislow et al. found that most women correctly classified the duration of lactation, even at the age of 69–70 years [45]. It is also believed that the metabolic health of a woman entering pregnancy may have an impact on the lactation outcomes. Therefore, low lactation efficiency may be a marker of future risk of maternal metabolic disease [46].

The strength of this study is primarily the large number of participants (*n* = 7621) from an ethnically homogenous population. Many confounding variables, which are known to be associated with the risk of MetS and its components, including smoking, physical activity, and dietary patterns, were taken into account in this analysis. The results obtained for the entire study population were verified by performing separate analyses involving parity category. Due to the large dataset, the results of this study make an important contribution to documenting the relationship between breastfeeding and women’s metabolic health later in life.

## 5. Conclusions

A total breastfeeding duration of more than 12 months significantly reduced the risk of MetS in the study population. Additional analyses performed in three parity categories showed that the strongest protective effect against MetS was found for lactation lasting for a maximum 18 months and in women with at least two children. Moreover, women with more than three children had lower odds of increased TG concentration, and those who breastfed for 1–6 months also had lower odds of abdominal obesity. Therefore, breastfeeding can be recommended to reduce the risk of MetS and its components in later life. Support from healthcare professionals and lactation support interventions can greatly influence breastfeeding rates [47,48]. In Poland, the proportion of mothers breastfeeding at 6 and 12 months is currently 38% and 17%, respectively [48]. Therefore, more effective breastfeeding counseling is required in Poland. Further studies are also needed to understand the mechanisms underlying the effect of lactation on women’s metabolic health in later life.

## Figures and Tables

**Table 1 nutrients-12-02691-t001:** Baseline characteristics of the study population (*n* = 7621).

Factors		Metabolic Syndrome		Abdominal Obesity		Increased Glucose Concentration		Elevated Blood Pressure		Increased Triglyceride Concentration		Decreased HDL-Cholesterol Concentration	
		Yes *n* = 3216	No *n* = 4405	Yes *n* = 5735	No *n* = 1886	Yes *n* = 2074	No *n* = 5547	Yes *n* = 5318	No *n* = 2303	Yes *n* = 2345	No *n* = 5276	Yes *n* = 2334	No *n* = 5287
Age (years), X ± SD		**57.1 ± 4.9**	**54.2** **± 5.3**	**56.0** **± 5.2**	**53.7** **± 5.4**	**56.8** **± 5.0**	**54.9** **± 5.4**	**56.2** **± 5.2**	**53.7** **± 5.3**	**57.1** **± 4.9**	**54.7** **± 5.4**	**56.9** **± 5.1**	**54.7** **± 5.3**
Years of education, X ± SD		**12.7 ± 3.1**	**13.7** **± 3.1**	**12.94** **± 3.14**	**14.22** **± 3.03**	**12.8** **± 3.1**	**13.4** **± 3.2**	**13.1** **± 3.2**	**13.7** **± 3.2**	**12.7** **± 3.1**	**13.5** **± 3.2**	**12.7** **± 3.1**	**13.5** **± 3.2**
Place of living, N;%	City	2018; 62.75	2766; 62.79	**1417;** **24.71**	**510;** **27.04**	1307; 63.02	3477; 62.68	3344; 62.88	1440; 62.53	1457; 62.13	3327; 63.09	1499; 64.22	3285; 62.13
	Village	1198; 37.25	1639; 37.21	**4318;** **75.29**	**1376;** **72.96**	767; 36.98	2070; 37.32	1974; 62.88	863; 37.47	888; 37.87	1949; 36.94	835; 35.78	2002; 37.87
Marital status, N;%	Single	**854;** **26.55**	**1073;** **24.36**	**3422;** **59.67**	**1362;** **72.22**	536; 25.84	1391; 25.08	1344; 25.27	583; 25.31	**628;** **26.78**	**1299;** **24.62**	618; 26.48	1309; 24.76
	Married	**2362;** **73.45**	**3332;** **75.64**	**2313;** **40.33**	**524;** **27.78**	1538; 74.16	4156; 74.92	3974; 74.73	1720; 74.69	**1717;** **73.22**	**3977;** **75.38**	1716; 73.52	3978; 75.24
Parity, N;%	0	**250;** **7.77**	**347;** **7.88**	**410;** **7.15**	**187;** **9.92**	**167;** **7.75**	**430;** **7.75**	424; 7.97	173; 7.51	193; 8.23	404; 7.66	191; 8.18	406; 7.68
	1	**458;** **14.24**	**719;** **16.32**	**788;** **13.15**	**389;** **20.63**	**893;** **16.10**	**893;** **16.10**	800; 15.04	377; 16.37	349; 14.88	828; 15.69	343; 14.70	834; 15.77
	2	**1480;** **46.02**	**2054;** **46.63**	**2641;** **46.05**	**893;** **47.35**	**942;** **45.42**	**2592;** **46.73**	2483; 46.69	1051; 45.64	1081; 46.10	2453; 46.49	1078; 46.19	2456; 46.45
	3 and more	**1028;** **31.97**	**1285;** **29.17**	**1896;** **33.06**	**417;** **22.11**	**681;** **32.84**	**1632;** **29.42**	1611; 30.30	702; 30.48	722; 30.79	1591; 30.16	722; 30.93	1591; 30.09
Breastfeeding status, N;%	Parous, Never breastfed	359; 11.16	408; 9.26	**586;** **10.22**	**181;** **9.60**	236; 11.38	531; 9.57	543; 10.21	224; 9.73	**262;** **11.17**	**505;** **9.57**	249; 10.67	518; 9.80
Breastfed	1–6 mths	935; 29.07	1349; 30.62	**1635;** **28.51**	**649;** **34.41**	604; 29.12	1680; 30.29	1621; 30.48	663; 28.79	**680;** **29.00**	**1604;** **30.40**	684; 29.31	1600; 30.26
	7–12 mths	699; 21.74	927; 21.04	**1266;** **22.07**	**360;** **19.09**	415; 20.01	1211; 21.83	1115; 20.97	511; 22.19	**524;** **22.35**	**1102;** **20.89**	504; 21.59	1122; 21.22
	13–18 mths	331; 10.29	486; 11.03	**632;** **11.02**	**185;** **9.81**	230; 11.09	587; 10.58	562; 10.57	255; 11.07	**234;** **9.98**	**583;** **11.05**	237; 10.15	580; 10.97
	>18 mths	642; 19.96	888; 20.16	**1206;** **21.03**	**324;** **17.18**	422; 20.35	1108; 19.97	1053; 19.80	477; 20.71	**452;** **19.28**	**1078;** **20.43**	469; 20.09	1061; 20.07
Menopausal status, N;%	Pre.	**580;** **18.03**	**1561;** **35.44**	**1398;** **24.38**	**743;** **39.40**	**396;** **19.09**	**1745;** **31.46**	**1287;** **24.20**	**854;** **37.08**	**403;** **17.19**	**1738;** **32.94**	**446;** **19.11**	**1695;** **32.06**
	Post.	**2636;** **34.59**	**2844;** **64.56**	**4337;** **75.62**	**1143;** **60.60**	**1678;** **80.91**	**3802;** **68.89**	**4031;** **75.80**	**1449;** **62.92**	**1942;** **82.81**	**3538;** **67.06**	**1888;** **80.89**	**3592;** **67.94**
Hormone therapy, N;%	Yes	**324;** **10.07**	**631;** **14.32**	**624;** **10.88**	**331;** **17.55**	**200;** **9.64**	**755;** **13.61**	**620;** **11.66**	**335;** **14.55**	**240;** **10.23**	**715;** **13.55**	**244;** **10.45**	**711;** **13.45**
	No	**2892;** **89.93**	**3774;** **85.68**	**5111;** **89.12**	**1555;** **82.45**	**1874;** **90.36**	**4792;** **86.39**	**4698;** **88.34**	**1968;** **85.45**	**2105;** **89.77**	**4561;** **86.45**	**2090;** **89.55**	**4576;** **86.55**
Smoking, N;%	Yes	1536; 47.76	2073; 47.0	**2671;** **46.57**	**938;** **49.73**	1013; 48.84	2596; 46.80	2480; 46.63	1129; 49.02	**1159;** **49.42**	**2450;** **46.44**	1098; 47.04	2511; 47.49
	No	1680; 52.24	2332; 52.94	**3064;** **53.43**	**948;** **50.27**	1061; 51.16	2951; 53.20	2838; 53.37	1174; 50.98)	**1186;** **50.58**	**2826;** **53.56)**	1236; 52.96	2776; 52.51)
Traditional-carbohydrate DP, N;%	T1	1050; 32.65	1488; 33.78	1912; 33.34	626; 33.19	705; 33.99	1833; 33.04	**1723;** **32.40**	**815;** **35.39**	**747;** **31.86**	**1791;** **33.95**	**725;** **31.06**	**1813;** **34.29**
	T2	1052; 32.71	1485; 33.71	1906; 33.23	631; 33.46	681; 32.84	1856; 33.46	**1806;** **33.96**	**731;** **31.74**	**759;** **32.37**	**1778;** **33.70**	**785;** **33.63**	**1752;** **33.14**
	T3	1114; 34.64	1432; 32.51	1917; 33.43	629; 33.35	688; 33.17	1858; 33.50	**1789;** **33.64**	**757;** **32.87**	**839;** **35.78**	**1707;** **32.35**	**824;** **35.30**	**1722;** **32.57)**
Healthy DP, N;%	T1	**1147;** **35.67**	**1391;** **31.58**	1934; 33.72	604; 32.03	691; 33.32	1747; 33.30	**1826;** **34.34**	**712;** **30.92**	**874;** **37.27**	**1664;** **31.54**	**865;** **37.06**	**1673;** **31.6**
	T2	**1067;** **33.18**	**1471;** **33.39**	1921; 33.50	617; 32.71	703; 33.90	1835; 33.08	**1772;** **33.32**	**766;** **33.26**	**776;** **33.09**	**1762;** **33.40**	**781;** **33.46**	**1757;** **33.23**
	T3	**1002;** **31.16**	**1543;** **35.03**	1880; 32.78	665; 35.26	680; 32.79	1865; 33.62	**1720;** **32.34**	**825;** **35.82**	**695;** **29.64**	**1850;** **24.28**	**688;** **29.48**	**1857;** **35.12**
Westernized DP, N;%	T1	1051; 32.68	1487; 33.76	1882; 32.82	656; 34.78	708; 34.14	1830; 32.99	1761; 33.11	777; 33.74	765; 32.62	1773; 33.61	768; 32.90	1770; 33.48
	T2	1090; 33.89	1447; 32.85	1916; 33.41	621; 32.93	698; 33.65	1839; 33.15	1811; 34.05	726; 31.52	802; 34.20	1735; 32.88	799; 34.23	1738; 32.87
	T3	1075; 33.43	1471; 33.39	1937; 33.78	609; 32.29	668; 32.21	1878; 33.86	1746; 32.83	800; 34.74	778; 33.18	1768; 33.18	767; 32.86	1779; 22.81
Physical activity (MET/min/day^−1^), X ± SD		**574.6** **± 460.9**	**621.1** **± 489.4**	597.1 ± 474.4	614.7 ± 488.1	**565.2** **± 464.1**	**615.1** **± 482.6**	**590.5** **± 470.4**	**627.0** **± 494.6**	**580.7** **± 461.7**	**610.7** **± 485.0**	**573.2** **± 467.6**	**614.0** **± 482.2**
Sitting time (min/day), X ± SD		**288.7** **± 130.3**	**299.4** **± 137.5**	**292.5** **± 132.4**	**301.8** **± 140.8**	**289.8** **± 131.3**	**296.7** **± 135.8**	293.1 ± 133.7	299.0 ± 136.5	**287.4** **± 129.0**	**298.1** **± 136.9**	**287.9** **± 13.1**	**297.9** **± 136.2**
BMI [kg/m^2^], X ± SD		**30.25** **± 4.92**	**26.28** **± 4.23**	**29.57** **± 4.51**	**23.05** **± 2.18**	**30.27** **± 5.35**	**27.09** **± 4.48)**	**28.86** **± 5.09**	**25.87** **± 3.85**	**29.49** **± 5.05**	**27.27** **± 4.73**	**29.60** **± 5.10**	**27.23** **± 4.69**

T1—the lowest; T2—from 33.3 to 66.6 percentile; T3—the highest; Pre.—premenopausal; Post.—postmenopausal; numbers in bold indicate statistically significant results.

**Table 2 nutrients-12-02691-t002:** Multivariable logistic regression analysis for metabolic syndrome (MetS) and its components in relation to breastfeeding duration (adjusted for age, BMI, education, place of living, marital status, parity, menopausal status, hormone therapy, physical activity, sitting time, traditional, healthy, westernized dietary patterns, smoking).

Breastfeeding Status	Metabolic Syndrome		Abdominal Obesity		Increased Glucose Concentration		Elevated Blood Pressure		Increased Triglyceride Concentration		Decreased HDL-Cholesterol Concentration	
	OR (95% CI)	*p*	OR (95% CI)	*p*	OR (95% CI)	*p*	OR (95% CI)	*p*	OR (95% CI)	*p*	OR (95% CI)	*p*
Never breastfed	1.00		1.00		1.00		1.00		1.00		1.00	
Ever breastfed	0.85 (0.71–1.003)	0.054	0.86 (0.66–1.11)	0.248	0.84 (0.70–1.001)	0.051	1.03 (0.86–1.23)	0.763	0.88 (0.75–1.05)	0.150	0.96 (0.81–1.14)	0.637
Breastfed 1–6 months	0.87 (0.72–1.04)	0.137	0.78 (0.61–1.04)	0.096	0.88 (0.73–1.07)	0.193	1.12 (0.93–1.36)	0.243	0.88 (0.73–1.05)	0.165	0.97 (0.81–1.16)	0.735
7–12 months	0.87 (0.71–1.06)	0.167	1.02 (0.76–1.37)	0.894	**0.77 (0.63–0.94)**	**0.012**	0.93 (0.76–1.15)	0.508	0.95 (0.78–1.16)	0.616	0.97 (0.81–1.16)	0.760
13–18 months	**0.76 (0.60–0.95)**	**0.017**	0.94 (0.66–1.33)	0.719	0.86 (0.68–1.10)	0.230	0.96 (0.76–1.22)	0.740	0.80 (0.64–1.01)	0.057	0.89 (0.70–1.18)	0.307
>18 months	**0.79 (0.64–0.98)**	**0.030**	0.80 (0.58–1.11)	0.181	0.81 (0.66–1.01)	0.063	0.96 (0.77–1.19)	0.722	0.85 (0.69–1.04)	0.119	0.97 (078–1.19)	0.746

Numbers in bold indicate statistically significant results.

**Table 3 nutrients-12-02691-t003:** Multivariable logistic regression analysis for MetS and its components in relation to parity and breastfeeding duration (adjusted for age, BMI, education, place of living, marital status, menopausal status, hormone therapy, physical activity, sitting time, traditional, healthy, westernized dietary patterns, smoking).

Parity	Breastfeeding Status	Metabolic Syndrome		Abdominal Obesity		Increased Glucose Concentration		Elevated Blood Pressure		Increased Triglyceride Concentration		Decreased HDL-Cholesterol Concentration	
		OR (95% CI)	*p*	OR (95% CI)	*p*	OR (95% CI)	*p*	OR (95% CI)	*p*	OR (95% CI)	*p*	OR (95% CI)	*p*
Nulliparous	-	1.00		1.00		1.00		1.00		1.00		1.00	
Parous	-	0.88 (0.72–1.07)	0.212	1.03 (0.77–1.37)	0.836	0.85 (0.69–1.05)	0.129	**0.81 (0.66–0.99)**	**0.048**	0.86 (0.71–1.04)	0.125	0.86 (0.71–1.05)	0.132
One child	Never	1.00		1.00		1.00		1.00		1.00		1.00	
	1–6 months	1.01 (0.75–1.37)	0.935	0.97 (0.71–1.32)	0.839	0.79 (0.57–1.11)	0.176	1.32 (0.97–1.80)	0.078	1.00 (0.73–1.38)	0.976	1.13 (0.82–1.56)	0.456
	7–12 months	1.00 (0.66–1.52)	0.996	1.17 (0.75–1.81)	0.483	0.73 (0.45–1.17)	0.191	0.90 (0.59–1.37)	0.631	1.07 (0.70–1.66)	0.748	1.03 (0.66–1.61)	0.884
	13–18 months	1.07 (0.51–2.27)	0.850	0.93 (0.45–1.89)	0.835	1.85 (0.88–3.88)	0.104	1.43 (0.68–3.01)	0.345	0.45 (0.18–1.16)	0.098	0.62 (0.25–1.51)	0.288
	>18 months	0.65 (0.31–1.35)	0.246	0.94 (0.49–1.82)	0.863	0.80 (0.36–1.79)	0.593	0.98 (0.51–1.14)	0.944	0.88 (0.42–1.87)	0.744	0.92 (0.43–1.96)	0.834
Two children	Never	1.00		1.00		1.00		1.00		1.00		1.00	
	1–6 months	**0.77 (0.60–0.97)**	**0.029**	0.76 (0.57–1.01)	0.054	0.83 (0.65–1.08)	0.162	0.93 (0.72–1.22)	0.616	0.84 (0.65–1.08)	0.167	0.82 (0.64–1.05)	0.109
	7–12 months	**0.75 (0.59–0.97)**	**0.026**	0.93 (0.69–1.25)	0.620	**0.75 (0.58–0.98)**	**0.038**	0.88 (0.67–1.15)	0.342	0.88 (0.67–1.14)	0.315	0.83 (0.64–1.08)	0.166
	13–18 months	**0.71 (0.52–0.96)**	**0.026**	0.79 (0.56–1.12)	0.191	0.84 (0.61–1.17)	0.301	0.86 (0.62–1.20)	0.378	0.89 (0.65–1.22)	0.475	0.88 (0.64–1.20)	0.420
	>18 months	0.86 (0.64–1.14)	0.289	0.99 (0.71–1.38)	0.947	0.94 (0.69–1.27)	0.680	0.99 (0.73–1.34)	0.935	0.92 (0.68–1.24)	0.580	0.90 (0.67–1.21)	0.496
Three or more children	Never	1.00		1.00		1.00		1.00		1.00		1.00	
	1–6 months	**0.60 (0.39–0.94)**	**0.026**	**0.42 (0.22–0.80)**	**0.008**	0.82 (0.52–1.28)	0.377	0.89 (0.54–1.47)	0.651	**0.62 (0.40–0.98)**	**0.042**	0.90 (0.57–1.43)	0.660
	7–12 months	0.77 (0.51–1.18)	0.232	0.62 (0.33–1.17)	0.142	0.68 (0.44–1.05)	0.079	0.73 (0.45–1.17)	0.190	0.82 (054–1.25)	0.353	1.01 (0.65–1.56)	0.966
	13–18 months	**0.65 (0.42–0.99)**	**0.050**	0.66 (0.35–1.27)	0.216	0.73 (0.47–1.14)	0.169	0.77 (0.48–1.26)	0.300	**0.61 (0.39–0.95)**	**0.029**	0.85 (0.54–1.33)	0.466
	>18 months	0.72 (0.48–1.07)	0.103	0.59 (0.32–1.10)	0.095	0.72 (0.48–1.09)	0.121	0.81 (0.52–1.28)	0.372	0.70 (0.47–1.05)	0.082	1.01 (0.67–1.54)	0.953

Numbers in bold indicate statistically significant results.

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
