# Peer review of "Breastfeeding and Prevalence of Metabolic Syndrome among Perimenopausal Women"

_nutrients, 2020, doi:10.3390/nu12092691_

Round 1
Reviewer 1 Report
I think that the issue of this study is very important and interesting. However, I read the manuscript and have some difficulties to follow the results.
Most of my remarks are technical:
I would suggest to present the results not in one long paragraph but in subtitles according to the massage want to be presented.
The tables are huge and difficult to follow and understand.
The main issue of the relationship between breastfeeding and MeS was lost in the text in the way it is presenting now.
So I recommend that the manuscript should be reorganized in such a form that it will easy to read and followed.
Another important remark- the presence of MeS in not only yes/no question. The number of MeS components is very important. Theoretically, there can be a difference between those with 2-3-4-5 MeS components. But it is not clear from the manuscript how many MeS components each individual had. I think that adding this analysis will help to increase the impact of the results.
Author Response
Response to the reviewers (Rev.1)
Thank you very much for the comments and suggestions about our manuscript entitled “Breastfeeding and prevalence of metabolic syndrome among perimenopausal women”.
Rev. 1
I would suggest to present the results not in one long paragraph but in subtitles according to the massage want to be presented.
In the revised manuscript section Results has been divided into subsections: 3.1. Characteristics of MetS, breastfeeding duration, sociodemographic, biomedical and lifestyle factors of the study participants (lines 141-142); 3.2. Analysis of the relationship between breastfeeding duration and the presence of MetS and its components (lines 189-190); 3.3. The probability of development of MetS and its components in relation to parity and breastfeeding duration (lines 206-207).
The tables are huge and difficult to follow and understand.
Table 1 has been improved.
The main issue of the relationship between breastfeeding and MeS was lost in the text in the way it is presenting now. So I recommend that the manuscript should be reorganized in such a form that it will easy to read and followed.
The manuscript has been corrected.
Another important remark- the presence of MeS in not only yes/no question. The number of MeS components is very important. Theoretically, there can be a difference between those with 2-3-4-5 MeS components. But it is not clear from the manuscript how many MeS components each individual had. I think that adding this analysis will help to increase the impact of the results.
We agree with the Reviewer that the number of MeS components is very important. However, we came to the conclusion that such additional analysis would increase the volume of the manuscript. It may possibly be the subject of a subsequent publication. In this manuscript, according to the IDF guidelines, MetS was diagnosed in the presence of 3 or more of the 5 risk factors. This information has been introduced in the revised manuscript, in lines 93-100.
Reviewer 2 Report
Protection against metabolic syndrome in later life is often cited as an important health benefit of breastfeeding for the mother, but studies have not found consistent associations between breastfeeding and women’s later metabolic health. This study utilises a large, robust dataset to add needed evidence to this body of literature. The stated aim of the study (investigate associations between breastfeeding duration and MetS and its components), however, seems limited for the analyses conducted. This makes it difficult for the reader to determine exactly how this study contributes to knowledge on this topic.
The authors are requested to more clearly describe the aims of their study in the introduction, and to revise the discussion to focus on the results of their study. Sections of the discussion, esp. ln 203-244, contain significant literature review that may be more appropriate as background information or could be more effectively linked to give context to the results of this study.
Specific comments are:
Ln 50-51 – Nguyen et al also note that this relationship should be interpreted with caution because the evidence comes from a small number of observational studies.
Ln 56-58 – The authors may consider revising sentences like this in the manuscript to improve readability, such as “breastfeeding for more than six months was associated with…”
Ln 62-64 – The aims of the study should be more fully and clearly described here. Including, the authors should specify the parity groups and if the analysis is only conducted on parous women. If nulliparous women are included, then the aim of the study would seem to be not only to determine the association of breastfeeding duration but also pregnancies on MetS and its components.
Ln 78-79 – It would be useful to provide a very brief summary of the components of MetS investigated as part of this study here, or in the introduction.
Ln 82 - Breastfeeding duration is summarised in Table 1 as Never, 1-6 months, 7-12 months, etc. Could women report breastfeeding <1 month? If so, please clarify how they were classified.
Ln 106 – 109 – Nulliparous women are not expected to have had any breastfeeding experience. It would be useful to clarify the aims of the study or to have further explanation as to why nulliparous women were included in these analyses if the goal of the study is to examine associations between breastfeeding duration and MetS.
Ln 117 - Descriptive subheadings in this section would be useful for organizing the presentation of results.
Table 1 – In ‘Breastfeeding status’, if ‘never breastfed’ includes nulliparous women this should be indicated or this line should be split into ‘Never breastfed, nulliparous’ and ‘Never breastfed, parous’.
Ln 200-201 – Replace ‘the relationships worsened for the general study population’ with more precise language.
Ln 203 – Given that the results of previous studies in this area are mixed, it would be useful to be more specific when the authors state ‘Our findings agree with the results of many long-term and cross-sectional studies.’
The authors may also be interested in Ra & Kim. Beneficial Effects of Breastfeeding on the Prevention of Metabolic Syndrome among Postmenopausal Women. Asian Nursing Research 2020; doi.org/10.1016/j.anr.2020.07.003
Ln 217-222 – This paragraph could be clarified to indicate that the authors are discussing the results of this study.
Ln 227, also Ln 266 – Please avoid ‘proved’ in this context, replace with ‘demonstrated’, ‘found’, etc.
Ln 260 – The authors should consider if maternal metabolic health factors before pregnancy that they cannot account for in this study might be relevant, see: Steube. Does breastfeeding prevent the metabolic syndrome, or does the metabolic syndrome prevent breastfeeding? Seminars in Pernatology 2015;39(4):260-295.
The discussion would also benefit from a summary of what this study adds to the current literature, and what still remains to be investigated.
Author Response
Response to the reviewers (Rev.2)
Thank you very much for the comments and suggestions about our manuscript entitled “Breastfeeding and prevalence of metabolic syndrome among perimenopausal women”.
Rev. 2
The authors are requested to more clearly describe the aims of their study in the introduction, and to revise the discussion to focus on the results of their study. Sections of the discussion, esp. ln 203-244, contain significant literature review that may be more appropriate as background information or could be more effectively linked to give context to the results of this study.
The aim of the study and fragments of the discussion have been changed in the revised manuscript. The aims were more clearly indicated in the introduction, lines 67-71. Discussion sections was modified in lines 245-247, 258, and 334-336.
Specific comments are:
Ln 50-51 – Nguyen et al also note that this relationship should be interpreted with caution because the evidence comes from a small number of observational studies.
It has been corrected in the revised manuscript in lines 53-55.
Ln 56-58 – The authors may consider revising sentences like this in the manuscript to improve readability, such as “breastfeeding for more than six months was associated with…”
It has been corrected in the revised manuscript in line 60.
Ln 62-64 – The aims of the study should be more fully and clearly described here. Including, the authors should specify the parity groups and if the analysis is only conducted on parous women. If nulliparous women are included, then the aim of the study would seem to be not only to determine the association of breastfeeding duration but also pregnancies on MetS and its components.
Corrected (lines 65-71).
Ln 78-79 – It would be useful to provide a very brief summary of the components of MetS investigated as part of this study here, or in the introduction.
Done (lines 93-100)
Ln 82 - Breastfeeding duration is summarised in Table 1 as Never, 1-6 months, 7-12 months, etc. Could women report breastfeeding <1 month? If so, please clarify how they were classified.
The division into breastfeeding categories was made on the basis of full months. <1 month category was not included, as clearly indicated in the revised manuscript, lines 106-107.
Ln 106 – 109 – Nulliparous women are not expected to have had any breastfeeding experience. It would be useful to clarify the aims of the study or to have further explanation as to why nulliparous women were included in these analyses if the goal of the study is to examine associations between breastfeeding duration and MetS.
The aims of the study have been corrected. Nulliparous women were included in these analyzes because the study also analyzed the risk of MetS in parous versus nulliparous women. Please see the revised Introduction section, lines 67-71.
Ln 117 - Descriptive subheadings in this section would be useful for organizing the presentation of results.
In the revised manuscript section Results has been divided into subsections 3.1. Characteristics of MetS, breastfeeding duration, sociodemographic, biomedical and lifestyle factors of the study participants (lines 141-142); 3.2. Analysis of the relationship between breastfeeding duration and the presence of MetS and its components (lines 189-190); 3.3. The probability of development of MetS and its components in relation to parity and breastfeeding duration (lines 206-207).
Table 1 – In ‘Breastfeeding status’, if ‘never breastfed’ includes nulliparous women this should be indicated or this line should be split into ‘Never breastfed, nulliparous’ and ‘Never breastfed, parous’.
corrected
Ln 200-201 – Replace ‘the relationships worsened for the general study population’ with more precise language.
Corrected (line 242)
Ln 203 – Given that the results of previous studies in this area are mixed, it would be useful to be more specific when the authors state ‘Our findings agree with the results of many long-term and cross-sectional studies.’
It has been corrected in the revised manuscript and the particular references has been provided in lines 245-246.
The authors may also be interested in Ra & Kim. Beneficial Effects of Breastfeeding on the Prevention of Metabolic Syndrome among Postmenopausal Women. Asian Nursing Research 2020; doi.org/10.1016/j.anr.2020.07.003
Thank you for the suggestion, the study has been mentioned and the reference was introduced in the revised manuscript in lines 280-282.
Ln 217-222 – This paragraph could be clarified to indicate that the authors are discussing the results of this study.
Done (line 258).
Ln 227, also Ln 266 – Please avoid ‘proved’ in this context, replace with ‘demonstrated’, ‘found’, etc.
It has been corrected in the revised manuscript. Please see lines 264 and 313.
Ln 260 – The authors should consider if maternal metabolic health factors before pregnancy that they cannot account for in this study might be relevant, see: Steube. Does breastfeeding prevent the metabolic syndrome, or does the metabolic syndrome prevent breastfeeding? Seminars in Pernatology 2015;39(4):260-295.
Thank you for the suggestion, the study has been mentioned and the reference was introduced in the revised manuscript in lines 327-329.
The discussion would also benefit from a summary of what this study adds to the current literature, and what still remains to be investigated.
A short summary has been added to the discussion (lines 334-336). What still remains to be investigated – lines 348-350.